# Ligament Alteration in Diabetes Mellitus

**DOI:** 10.3390/jcm11195719

**Published:** 2022-09-27

**Authors:** Olga Adamska, Artur Stolarczyk, Agata Gondek, Bartosz Maciąg, Jakub Świderek, Paweł Czuchaj, Krzysztof Modzelewski

**Affiliations:** 1Orthopaedic and Rehabilitation Department, Medical University of Warsaw, 02-091 Warsaw, Poland; 2Department of Methodology, Medical University of Warsaw, 02-091 Warsaw, Poland; 3Faculty of Medicine, Medical University of Bialystok, 15-089 Bialystok, Poland

**Keywords:** diabetes mellitus, ligaments alteration, AGEs, ROS, connective tissue, diabetes mellitus complications, diabetes ligaments, comorbidities, orthopaedic surgery

## Abstract

Connective tissue ageing is accelerated by the progressive accumulation of advanced glycation end products (AGEs). The formation of AGEs is characteristic for diabetes mellitus (DM) progression and affects only specific proteins with relatively long half-lives. This is the case of fibrillar collagens that are highly susceptible to glycation. While collagen provides a framework for plenty of organs, the local homeostasis of specific tissues is indirectly affected by glycation. Among the many age- and diabetes-related morphological changes affecting human connective tissues, there is concurrently reduced healing capacity, flexibility, and quality among ligaments, tendons, bones, and skin. Although DM provokes a wide range of known clinical disorders, the exact mechanisms of connective tissue alteration are still being investigated. Most of them rely on animal models in order to conclude the patterns of damage. Further research and more well-designed large-cohort studies need to be conducted in order to answer the issue concerning the involvement of ligaments in diabetes-related complications. In the following manuscript, we present the results from experiments discovering specific molecules that are engaged in the degenerative process of connective tissue alteration. This review is intended to provide the report and sum up the investigations described in the literature concerning the topic of ligament alteration in DM, which, even though significantly decreasing the quality of life, do not play a major role in research.

## 1. Introduction

A lack of physiological action of insulin in diabetes mellitus (DM) induces an impaired metabolism of carbohydrates, lipids, and proteins, which are the necessary components for cellular homeostasis and tissue activities [1]. Altered glucose metabolism impacts on all the basic processes taking place in the organism and remains the reason for impaired body functioning and regenerative abilities [2]. DM is a complex metabolic disorder, which has several direct and indirect effects on multiple processes, starting from simple to more complex: chemotaxis, phagocytosis, bacterial killing, heat shock, protein expression, antioxidant synthesis, oxygen-free-radical generation, growth factor depletion, glucocorticoid concentration, cell proliferation, up-regulation of apoptosis, and extracellular matrix (ECM) synthesis [3,4]. The cellular and molecular background for those changes and irreversible degradation are further evaluated and, so far, the pathophysiology of tendinopathies has been clarified comprehensively. Despite the ligaments remaining frequently injured in patients in whom insulin resistance or DM developed and the fact that they are the key structures in joint balance, the influence of body hyperglycaemia is poorly understood [5].

Ligaments are compositions of collagenous tissue that create the joints and link bones together. Two-thirds of the inner biochemical content of ligaments is water, and one-third consists of solid components. Those in the majority consist of collagen (type I collagen accounting for 85% of the collagen), proteoglycans, elastin and other proteins, and glycoproteins such as actin, laminin, and integrins [6]. Ligaments play a crucial role in the motoric system by responding to loads and micro-injuries that affect them during the whole lifetime with an increased mass and stiffness. All the life activities together with ageing, maturation, tension, and exercise given to the joint affect the biomechanical properties of ligaments. While they play a key role in functioning and weight bearing, there is clinical evidence of alteration of their properties caused by DM, but the current knowledge is not comprehensive. Joint degenerative and inflammatory diseases such as commonly diagnosed osteoarthritis (OA), pending pain, and reduced activities of daily living are the top complaints of patient who attend orthopaedic outpatient clinics. The pain that accompanies the acceleration of pro-inflammatory molecules is the number-one cause of disability worldwide [7,8,9,10]. DM, on the other hand, will likely affect 500 million people worldwide by 2030 [8]. Diabetes is the seventh leading cause of death [9], while DM and OA affect about 1 billion people worldwide [10]. Both remain a serious public health issue; therefore, they require extensive investigation into their mutual dependencies.

In the present study, we aimed to research the mechanism responsible for ligament disorders and malfunctioning in patients suffering from DM, based on the currently available literature, as well as to conclude the fields for prospective research that must be conducted to illuminate the concern.

## 2. The Formation of Advanced Glycation End Products (AGEs) Is Considered to Be the Biggest Risk Factor for the Development of Diabetic Complications

Persistent hyperglycaemia provokes glucose interference with plasma molecules, covalent formations with lipids, and proteins via non-enzymatic glycation, which determines the alteration of molecular mechanisms and the overall tissue condition [10].

Protein glycation and the formation of AGEs play a crucial role in the pathogenesis of diabetic complications. AGEs create intra- and extracellular cross-links with proteins, sometimes lipids and nucleic acids, which participate in the progression of diabetic complications. They cause disruption in the molecular conformation, rearrange enzymatic activity, and disorganise molecule-receptor functioning [4,11,12]. The investigation of cellular receptors for AGEs (RAGE) revealed the pathophysiological bases for the alteration of intracellular signalling as well as expression of important genes, pro-inflammatory molecules, and free-radical release [6]. The activated RAGE promotes the chain reaction of reactive oxygen species and triggers the transcription of factor NF-κB [4,13]. Cross-link formation in collagen fibres dictates the development of pro-inflammatory molecules, which then accelerate the creation of a disorganised cell phenotype [14,15,16].

## 3. Negative Impact of Diabetes on the Musculoskeletal System

Beyond the most commonly occurring complications of DM, musculoskeletal disorders are commonly found, receiving unproportionally little attention in general. The evidence shows that musculoskeletal injuries in individuals with DM may occur due to the AGEs, increasing collagen crosslinking and generating abnormal collagen deposits in tendons and ligaments as well as abnormal thickening and joint rigidity [15,16]. Research concerning orthopaedic problems of patients with DM remains scarce, but musculoskeletal disabilities provoke substantial burdens on quality of life. When vascular and neurological deficits appear, they underline the prudence of the control of a glycaemic state in orthopaedics in terms of recovery [17]. Figure 1 presents possible complications of DM on orthopaedic patients.

Since 12% of all patients visiting orthopaedists suffer from DM, it is important to order the current knowledge about its effects on the musculoskeletal system [18].

A study conducted by Frisch et al. reported the connection between DM in patients undergoing surgery and an elevated complication ratio due to the poor quality of soft tissues, followed by prolonged hospital stays and higher perioperative mortality [19].

In general, surgery induces metabolic stress, catabolic hormone secretion, and breaking of an anabolic hormone action such as insulin. Patients with DM, who do not have an impaired insulin secretion and are treated with a major surgery, have an even more exacerbated metabolic stress-related response for increased insulin demand. Major surgery provokes functional insulin insufficiency [20]; therefore, diabetic patients are at higher risk for a procedure-induced infection and poor glycaemic control [19,21,22].

Studies consistent support that well-controlled DM is key for a successful surgery. Management by a multidisciplinary team and attention to discharge planning are the key aspects of care during and after orthopaedic surgery performed on patients with metabolic disorders [20,21,22]. However, there are currently no guidelines for orthopaedic departments to provide patients with a holistic approach to their metabolic disorders.

## 4. Heterogeneity in Connective Tissue Treatment Modalities—Tendons and Ligaments

It is generally recognized among the population worldwide that ageing societies carry out less physical activity. Additionally, joint stiffness is provoked by increased quantities of collagen cross-linking, an effect of AGEs. Due to differential histological formation between tendons and ligaments, the pathophysiologies of both tissues are heterogeneous. Furthermore, the prognosis for ligament healing is much worse [23,24].

Despite ligaments and tendons being functionally and grossly similar, mainly composed of collagen type I (Col1A1) and collagen type 3 (Col3A1), they still present as heterogeneous according to histological and biochemical characteristics. Plenty of studies concern tendon alteration facilitated by DM, but the literature is lacking regarding ligament redaction. The differences in tissue-specific collagen-maturation processes in fibroblasts creating ligaments and tendons are still unknown, and they provide the reason for the inferiority of ligaments in terms of collagen synthesis, proliferation, and migration. Studies found it to be a crucial aspect of the relatively poor healing potential of ligaments in comparison to tendons [21,22].

Experimental studies relying on animal models compared the morphological and biochemical features of ligaments and tendons and revealed a higher ratio of enzymatic content of the lysine hydroxylase 2/lysine hydroxylase 1 in ligaments. The expression of lysyl oxidase has a regulatory effect on the amount of enzymatic cross-linking. The levels of Col1A1 and Col3A1 were additionally greater in the case of the ligament matrices than in the tendon matrices. Ligament- and tendon-derived cells have distinct collagen-maturation processes at the cellular level, and collagen maturation of ligament cells is not necessarily inferior to that of tendons with regard to collagen synthesis and maturation [23].

The literature shows the decreased proliferation rate, higher turnover, and breakdown of fibroblasts derived from soft tissues induced by DM. These suggest that excessive proliferation together with an altered structure of the fibroblasts may contribute to a poor production of collagen [24,25]. Further investigations evidence that DM fibroblasts have impaired migration and phenotypic change, increase in matrix metalloproteinase 9 (MMP-9), and diminished production of vascular endothelial growth factor (VEGF) [26,27,28].

## 5. Rheumatoid Arthritis (RA) and T1DM

It has long been analysed that people with RA have a higher risk of T1D, and vice versa, but recent studies have concluded that the association of RA and T1D appears to be limited and specific to those RA patients with positive anti-citrullinated peptide antibodies. The risk of developing RA in later life was attributed partly to the presence of a specific allele 620 W PTPN22, possibly representing a common pathway for both autoimmune diseases. To date, both animal and human studies have yielded conflicting and inconsistent results linking DM with OA initiation and progression, and more rigorous data are needed [11,12]. Hyperglycaemia induces pro-inflammatory mediators, local toxicity, hypertrophy, and apoptosis in soft tissues of the joints [29]. DM induces chondrocytes to become hypertrophic and produce catabolic factors such as Interleukin 1ß and Fibroblast Growth Factor 2, which increases ECM degradation [29,30]. They also facilitate the further production of inflammatory mediators such as nitric oxide, prostaglandin E2, and ROS.

The study conducted by Njoto I. et al. confirms that DM with characteristic perlecan expression is a trigger for OA [30,31]. Radojčić MR et al. revealed that there are associations between the novel ECM biomarker C1M and local and systemic interleukin 6 with synovitis and pain, finding a net loss of collagen elevation and increase in levels of MMP-2, -9, and -13. Furthermore, cathepsin K is responsible for the degradation of C1M, and thus, the osteoclast’s main protease, which enables its release from bone. It is a soft connective-tissue-degradation biomarker found to be released in an ex vivo model of synovitis as well as in RA, ankylosing spondylitis, and osteoarthritis. Inflammation then involves the majority of cytokines, which are essential in cellular communication and important mediators in the aberrant metabolism of articular structures [32].

## 6. Manifestation of Ligaments Pathologies in Diabetic Patients—Specific Findings

Studies link a higher prevalence of hypertrophy of ligamentum flavum (LF) to DM. It contributes to lumbar spinal stenosis (LSS), including fibrocartilaginous pathological degeneration due to the excessive proliferation of collagen, calcium crystal deposition, collagen and elastin fibre changes, and metaplasia of the ligament fibroblasts. With age, the elastin-to-collagen ratio decreases, resulting in decreased elasticity and increased stiffness or fibrosis. In DM, LF degradation is even more prominent, and it is caused through the enzymatic degradation of ECM and tissue remodelling by MMPs, TIMPs, platelet-derived growth factor-BB, connective tissue growth factor, bone morphogenetic protein (BMP), and inflammatory cytokines [33,34,35,36]. Specimens of the LSS patients presented with increased infiltration of inflammatory cells and were stained positively for MMP-3, MMP-9, vimentin, fibronectin, and increased ROS [35,36,37]. Shemesh S. et al. revealed enhanced elastin fibre losses in DM patients and the positive correlation of this loss to fasting plasma glucose.

The changes that occur in the extracellular matrix of diabetic patients’ ligaments may differ from the normal ageing process in two important ways: an increased nonenzymatic cross-link of proteins by sugar glycosylation at lysine residues and a decreased rate of proteoglycan synthesis. Only a few studies linked diabetes and LSS [33,34,35]. Other studies demonstrated a protective effect of spinal decompression surgery on diabetic patients with LSS, since successful surgery may improve the level of physical activity and thus facilitate glycaemic control [38,39]. Nonetheless, the literature demands further histological investigations to confirm the contribution of LF hypertrophy to DM [2].

Lin et al. found that culturing rat connective tissue in hyperglycaemia for up to 48 h not only causes decreases in the expression of Col1A1 but also scleraxis and tenomodulin, as well as apoptosis and decreased proliferation [40]. Wu et al. disagreed with the aforementioned but confirmed the exacerbation of the expression of genes including mohawk, biglycan, and transforming growth factor β-1 [41]. These studies suggest that the connective tissue is significantly affected. The increase in metabolic stress, propensity to undergo apoptosis, and behaviour in situations of oxidative stress are all complications of a hyperglycaemic state in the organism over time [42].

Diabetes-induced cytokine release activates the JAK/STAT pathway that is responsible for development of the majority of cell types, including chondrocytes, osteocytes, and fibrocytes. An error occurring in order to the local or systemic inflammation causes an immediate response in disturbed signalling in this tightly regulated process [43]. Damerau A. et al. observed that inflammatory-induced JAK/STAT signalling stimulates osteoclasts, which results in an enhanced bone resorption [44]. Furthermore, it contributes to the production of matrix-damaging enzymes within the synovial fluid, which leads to cartilage and ligament destruction and bone erosion in late stages [43,44].

## 7. Histological and Biochemical Changes in Ligaments Specifically in Diabetic Patients

The process of ageing facilitates morphological and histological changes in the appearance of connective tissue. The glistening white appearance of connective tissue under the microscope is not recognizable anymore at a certain extent of ageing. DM was found to be a cofactor for exacerbating this change even more abruptly. Severely affected fragments show up greyish and amorphous, disproportionally distinguished into fusiform or nodular thickening portions [45].

Histopathological screening reveals a decreased number of fibroblasts and tenocytes together with a significantly decreased number of collagen fibres. Furthermore, they seem disintegrated and frayed, and their elasticity is consistent. The prognosis is rather poor due to the fact that detection of the condition usually takes place when the ligaments and tendons are already significantly thickened. In such an advanced stage, the normal daily-life functioning remains impaired and both the quantitative and qualitative features of the tissue are irreversibly decreased. Moreover, disturbed blood flow as a result of the progression of DM disorganizes the fundamental components of the tissue, which hinders the recovery [46]. Investigation of the molecular and cellular pathophysiology of ligaments must be prioritised in order to achieve satisfactory remission. The current findings of mechanisms altered specifically in ligaments are summarised in Table 1.

Tan J et al. found that, in an in vivo-induced hyperglycaemic microenvironment, high glucose inhibits the osteogenic differentiation of the periodontal ligament (PDL), thus inhibiting the synthesis and secretion of the collagen matrix and calcium salts to an adequate degree [47]. Endoplasmic reticulum (ER) stress activated by overstimulation mainly involves three unfolded protein-response signalling pathways, including activating transcription factor 6 (ATF6) pathway, double-stranded RNA-activated protein kinase (PKR)-like endoplasmic reticulum kinase (PERK) pathway, and inositol-requiring enzyme 1 (IRE1) pathway. While moderate ER stress can effectively protect the body, excessive ER stress causes degeneration [48,49]. Song X et al. evidenced that ER causes apoptosis in PDL and vascular calcification in a rat model [50]. In the previous study of Tan J et al., ER stress was also found to reduce the osteogenic differentiation ability of PDL when influenced by tumour necrosis factor-α [46]. Both indicate that an inflammatory microenvironment causes drastic interruption throughout the physiological functioning of cell mechanisms in ligaments [49,51].

The balance of MMPs and tissue inhibitors of MMPs (TIMPs) is crucial for the stabilisation of the ECM, with an MMP/TIMP imbalance associated with the pathologic breakdown of the ECM [28,51,52,53]. In diabetes, collagen is rapidly degraded by elevated MMP levels and the fibroblast function is disturbed [54,55,56].

**Table 1 jcm-11-05719-t001:** Characteristics of studies on properties of ligament alterations among diabetic populations.

Literature	Species Model	Groups	Duration of DM	Analysed Tissue	Found Correlations
Li K. et al., 1995 [27]	Sprague–Dawley rats	SCG (*n* = 22) DM (*n* = 28) DM-IT (*n* = 7)	1 week	Ligament (MCL)	The length, thickness, and cross-sectional area of the DM MCL were significantly smaller than the control values—consistent with the reduced quantities of collagen in ligaments. MCL cell density was smaller in DM group compared with DM-IT, but DM-IT showed improvement in properties compared with untreated DM.
Vincente A. et al., 2020 [28]	Sprague–Dawley rats	CG (*n* = 20) DM (*n* = 20) DM-IT (*n* = 20)	11 days	Ligament (PDL)	Force applied to PDL of DM rats caused higher inflammatory response, more oxidative stress, and a greater extent of orthodontic tooth movement than in normoglycemic rats. Stress produces a greater disorganisation of PDL in diabetic rats together with higher MMP-8 and MMP-9 expressions. Greater expression of it is observed in diabetic patients, which leads to increased collagen and gelatine degradation. This provokes poor regenerative features and worse prognosis of mechanical recovery after trauma and mechanical stress.
Njoto I. et al., 2018 [30]	Rattus norvegicus strain Wistar	CG (*n* = 3) DM (*n* = 15)	61 days	Cartilage (chondrocytes and pericellular matrix)	Increases in glycemia of animal models interfere with chondrocyte shape and formation. Hyperglycaemia provokes production of pro-inflammatory mediators, such as AGEs, local toxicity to joint tissues, and apoptosis.
Njoto I. et al., 2019 [31]	Rattus norvegicus strain Wistar	CG (*n* = 5) DM (*n* = 15) DM-IT (*n* = 15)	21 days; 28 days; 42 days	Ligament (ACL)	Protein expression of perlecan in ligaments gradually decreased over time within DM groups. Hyperglycaemia predisposes articular cartilage damage, higher severity of the osteoarthritis disease, and reaches into the intracellular compartment.
Xin L. et al., 2010 [57]	Sprague–Dawley rats	CG (*n* = 24) DM (*n* = 24)	8 weeks	Ligament (PDL)	The DM group showed increased expression of MMP-1 and Col-III and decreased expression of Col-I in PDL. The DM group appeared to have worse recovery from damage caused by orthodontic movement. DM showed alterations in immune response, inflammation, extracellular matrix synthesis, and collagen destruction.
Tan J. et al., 2022 [47]	Genetically diabetic C57BLKS/J-*Lepr^db^* (*db*/*db*) mice and their C57BLKS/J wild-type littermates	DM (*n* = 10) IG (*n* = 10) CG (*n* = 8)		Ligament (PDL)	The mRNA expression levels of GRP78, ATF6, PERK, and XBP1 were highest in DM, followed by IG, and the lowest in CG. Hyperglycaemia activates ER stress. DM and IG microscopic observations showed disorganised cell arrangement in PDL, necrotic tissue, inflammatory cells, inflammation, granulation tissue hyperplasia, and disordered fibroblasts.
Tang L. et al., 2022 [58]	Genetically diabetic C57BLKS/J-Leprdb (*db/db*) mice and their C57BLKS/J wild-type littermates	DM (*n* = ?) CG (*n* = ?)	8 weeks	Ligament (PDL)	DM produced ROS with an increased MDA level indicating lipid peroxidation. SOD and GSH-Px levels, which refer to essential antioxidative scavengers of ROS, were significantly decreased in the serum of DM. The intracellular DNA damage measurement occurred based on markedly increased 8-OHdG expression in DM. The telomere oxidative damage (accelerated telomere shortening) was detected through an expression of 53BP1 and the colocalization of 53BP1 and TRF2 increase in the PDL of the DM group.
Li H. et al., 2008 [59]	Sprague–Dawley rats	DM (*n* = ?) CG (*n* = ?)	12 weeks	Ligament (Posterior longitudinal ligament tissues of cervical spine)	Hyperglycaemia increases the gene expression and protein synthesis of collagen types I and III, particularly in cells of the posterior longitudinal ligament.

## 8. Conclusions

A wide range of studies investigating the influence of diabetes on tendons in animal models have been conducted. Although tendons and ligaments differ significantly from their hierarchical structure, which lets them adapt to their function, there are also many similarities between the two tissues. There are no preclinical or clinical data focusing on biomechanical and histological changes in ligament properties on individuals with DM, as is evidenced in the case of tendons. While the cellular and molecular mechanisms of these alterations are still ambiguous, further well-designed clinical studies are needed to establish which diabetes-induced molecules influence the ligamentous fibres, causing their degeneration.

According to the conducted studies, the correlation of AGEs, excessive induction of MMPs release, formation of reactive oxygen species, and accumulation of ECM inflammatory cytokines seem to play a crucial role in complications of diabetic ligaments. Diabetes-induced ligament alteration differs from normal ageing and occurs in the ECM within two patterns: (1) an increased nonenzymatic cross-link of proteins by glycation at lysine residues and (2) a decreased rate of proteoglycan synthesis. Diabetic alterations of ligaments may especially originate from excessive stimulation and interaction with PPARs, JAK/STAT pathways, ER stress, and ROS and AGEs accumulation, which remains the speculated topic for future studies.

The present studies point to the contribution of DM to the loss of elastin fibres that occurs in ligament complications following DM. To date, the majority of original studies are obsolete and contribute only to LF and PDL. Hence, further research concerning knee and hip ligaments is required to detect alterations of the aforementioned mechanisms and molecules in joints that are frequently injured in OA. 

## Figures and Tables

**Figure 1 jcm-11-05719-f001:**
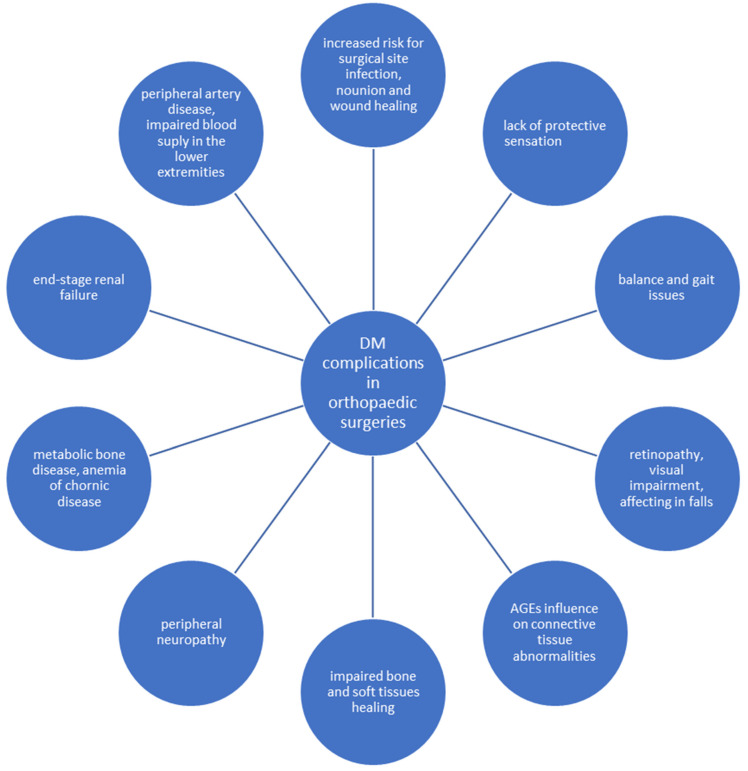
Potential impact of DM complications on patients undergoing orthopaedic surgeries [8].

## Data Availability

Data are available upon a request from author O.A.

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
