# Peer review of "Ligament Alteration in Diabetes Mellitus"

_jcm, 2022, doi:10.3390/jcm11195719_

Round 1

Reviewer 1 Report

It is a concise review of a rather specific complication of diabetes - DM tendinopathy.

I recommend a shortening of the title, such as Ligaments alteration in diabetes, since the Endoplasmic Reticular Stress, Inflammatory Cellular Fibroblast Dysfunction and Apoptosis is not the core of the manuscript.
The whole text should have English proof by a native speaker.
Figure 1 should be of better resolution and use a larger font.

Finally, several recommendations for possible future research on ligaments in diabetes should be mentioned.

Author Response

Dear Reviewer,

Thank your very much for your contribution to enhance the scientific value of the presenting manuscript.

I have added your comments and suggestions, including the change of the title, usage of a larger font in the diagram (figure 1.).

Finally, I have underlined in the ‘conclusions’ the specific directions of future research.

Also, the reference became updated as much as it is possible. The significant articles remained evet though, they are outdated, but the topic lacks on current interpretation and they conduct on the topic of the review.

The native speaker approval was also obtained for the English language.

Please, read again the introduced changes into the whole manuscript.

Thank you once again.

With kind regards,

The Authors.

Reviewer 2 Report

In general, this manuscript is well-organized and carefully reviewed the current concepts about the effect of diabetes mellitus to the connective tissue and its associated clinical musculoskeletal problems. But, there are still some comments listed below:

Although the word and abbreviation of “….advanced glycation end-products (AGE)…diabetes mellitus (DM)…” had been mentioned in the section of “ABSTRACT”, this statement should be re-mentioned in the main text of first appearance of the “INTRODUCTION” section.

Page 3

In the third section of this manuscript; it seems to me the subtitle should be change to: “ 3. Negative impact of diabetes in musculo-skeletal system “ in stead of the original one: “ 3. Negative impact of diabetes in orthopaedics”

Author Response

Dear Reviewer,

Thank your very much for your contribution to enhance the scientific value of the presenting manuscript.

I have added your comments and suggestions, including the abbreviations meaning in ‘’Introduction’’ section.

Additionally, I have used a larger font in the diagram (figure 1.) and changed the subtitle 3.

Finally, I have underlined in the ‘conclusions’ the specific directions of future research.

Also, the reference became updated as much as it is possible. The significant articles remained evet though, they are outdated, but the topic lacks on current interpretation and they conduct on the topic of the review.

The native speaker approval was also obtained for the English language.

Please, read again the introduced changes into the whole manuscript.

Thank you once again.

With kind regards,

The Authors.

Round 2

Reviewer 2 Report

All my questions were adequately responded.

Thank you.